# Impact of Depression and/or Anxiety on Mortality in Women with Gynecologic Cancers: A Nationwide Retrospective Cohort Study

**DOI:** 10.3390/healthcare13151904

**Published:** 2025-08-05

**Authors:** Yung-Taek Ouh, Eun-Yeob Kim, Nam Kyeong Kim, Nak-Woo Lee, Kyung-Jin Min

**Affiliations:** 1Department of Obstetrics and Gynecology, Korea University Ansan Hospital, Ansan-si 15355, Gyeonggi-do, Republic of Korea; oytjjang@gmail.com (Y.-T.O.); 54305knk@gmail.com (N.K.K.); nwlee@korea.ac.kr (N.-W.L.); 2Department of Medical Science Research Center, Korea University Ansan Hospital, Ansan-si 15355, Gyeonggi-do, Republic of Korea; key0227@korea.ac.kr

**Keywords:** gynecologic cancers, depression, anxiety disorders, Korean National Health Insurance Service (NHIS)

## Abstract

Objective: This study aimed to investigate the impact of depression and anxiety disorders on mortality in women diagnosed with gynecologic cancers, utilizing nationwide retrospective cohort data. Methods: Data from the Korean National Health Insurance Service (NHIS) database, covering women diagnosed with cervical, endometrial, or ovarian cancers between 2007 and 2014, were analyzed. Women diagnosed with depression or anxiety disorders within one year after cancer diagnosis were identified and compared with a control group comprising patients with gynecologic cancers who did not develop either disorder during the same post-diagnosis period. Mortality was evaluated as the primary outcome. Results: Of 85,327 women analyzed, 784 (0.9%) were diagnosed with depression or anxiety disorders. Patients with depression or anxiety exhibited significantly higher mortality (38.4% vs. 29.9%; *p* < 0.001). Multivariate analysis indicated that depression significantly increased mortality risk (OR 1.46, 95% CI 1.27–1.66), whereas anxiety alone showed no significant effect (OR 0.97, 95% CI 0.74–1.27). Combined depression and anxiety showed the highest mortality risk (OR 1.47, 95% CI 1.31–1.65). Conclusions: Depression and anxiety disorders are significant predictors of increased mortality in women with gynecologic cancers, emphasizing the necessity for integrated mental health assessment and interventions in oncologic care to improve both survival and quality of life.

## 1. Introduction

Gynecologic cancers, including cervical, endometrial, and ovarian cancers, represent significant global health challenges characterized by substantial morbidity and mortality [1]. The prognosis of these cancers can be influenced by multiple factors, including biological, clinical, and psychosocial determinants [2]. Among these, psychological distress, particularly depression and anxiety disorders, has gained increasing recognition as a crucial factor that affects cancer prognosis [3]. In addition to affecting prognosis, psychological distress may also influence treatment response. Emotional states such as depression and anxiety can alter pharmacokinetics and pharmacodynamics by modulating the HPA axis, immune function, and inflammatory pathways, ultimately affecting drug metabolism, efficacy, and tolerance. This underscores the importance of incorporating psychological evaluation into comprehensive cancer care.

Patients diagnosed with cancer frequently experience heightened psychological distress, with depression and anxiety disorders being among the most common mental health concerns [4]. Depression, marked by persistent sadness and impaired functioning, and anxiety, characterized by excessive worry and physiological stress responses, can significantly affect a patient’s quality of life and adherence to medical treatments [5]. Notably, depression and anxiety are not merely reactive conditions; they are increasingly recognized as independent risk factors for adverse clinical outcomes, potentially exacerbating cancer progression [6,7].

Several mechanisms have been proposed to explain how depression and anxiety might influence cancer prognosis negatively [8]. Chronic psychological stress can disrupt the hypothalamic–pituitary–adrenal (HPA) axis, leading to altered cortisol secretion patterns and heightened inflammatory responses, both of which can accelerate tumor growth and metastasis [7]. Additionally, these mental health disorders can negatively affect patient behavior, reducing adherence to medical regimens, decreasing participation in health-promoting activities, and delaying necessary medical care [9].

Previous studies have demonstrated associations between depressive symptoms and increased mortality in various cancers such as breast, colorectal, and lung cancers [10,11]. However, evidence specifically concerning gynecologic cancers remains limited and inconsistent [12,13]. Studies addressing psychological factors in gynecologic oncology have primarily been smaller-scale and hospital-based, limiting generalizability and robustness [14]. This underscores the need for extensive, population-based research to clarify the relationship between mental health disorders and gynecologic cancer outcomes comprehensively. Such investigations can provide crucial insights for developing integrated care strategies and enhance the quality of life and survival outcomes for these patients [15,16].

To address this knowledge gap with greater statistical power and generalizability, we conducted a nationwide retrospective cohort study leveraging the comprehensive database of the Korean National Health Insurance Service (NHIS). This study aimed to evaluate the association between post-diagnosis depression and anxiety disorders and mortality among women with gynecologic cancers, thereby addressing a significant knowledge gap in gynecologic oncology and mental health research.

## 2. Materials and Methods

### 2.1. Study Design and Population

We conducted a retrospective cohort study using data extracted from the Korean National Health Insurance Service (NHIS) database to investigate the association between depression, anxiety disorders, and mortality among women diagnosed with gynecologic cancers, including cervical, endometrial, and ovarian cancer, between 2007 and 2014. The inclusion criteria were as follows: (1) women with a confirmed diagnosis of one of the three specified gynecologic cancers during the study period, and (2) availability of at least one year of follow-up data following the cancer diagnosis. Exclusion criteria included a documented diagnosis of depression or anxiety disorder within one year prior to the cancer diagnosis, as well as missing data related to mental health status or mortality outcomes.

The study population was recruited from all administrative regions of South Korea, including major metropolitan cities (e.g., Seoul, Busan, Daegu, Daejeon, Incheon, Gwangju, Ulsan) and provinces (e.g., Gyeonggi-do, Gangwon-do, Chungcheong, Jeolla, Gyeongsang, and Jeju Island). Although regional information was available through place of residence, subgroup analyses by location were not conducted due to the scope and statistical power limitations of this study. However, regional variables were included in multivariate models to adjust for geographic variation. Future studies may further explore regional and environmental influences—including urbanization, climate, and socioeconomic context—on mental health outcomes in this population.

After applying these criteria, a total of 85,327 women were included in the final cohort: 784 patients in the depression/anxiety group (612 with depression and 172 with anxiety disorders), and 84,543 in the control group, defined as patients who did not receive a diagnosis of depression or anxiety disorder within one year following their cancer diagnosis. The cohort selection process is illustrated in Figure 1.

### 2.2. Identification of Depression and Anxiety Disorders

Depression and anxiety disorders were identified using the International Classification of Diseases, 10th Revision (ICD-10) codes. Patients were classified into the depression or anxiety disorder groups if they had at least two outpatient or inpatient claims within one year following their gynecologic cancer diagnosis.

### 2.3. Data Collection and Variables

Demographic and clinical variables collected included age, height, weight, Body Mass Index (BMI), health insurance premium, place of residence, days of hospitalization, surgery status, duration of care, and various clinical parameters such as blood pressure, AST (SGOT), ALT (SGPT), Gamma-glutamyl transferase (GGT), urine pH, hemoglobin, fasting blood glucose, total cholesterol, waist circumference, triglycerides, HDL cholesterol, LDL cholesterol, serum creatinine, and estimated glomerular filtration rate (eGFR). Cognitive function was evaluated using scores derived from the Korean version of the Mini-Mental State Examination (K-MMSE), where higher scores indicate poorer cognitive performance in this dataset. Physical activity levels were assessed based on the self-reported weekly frequency of vigorous, moderate, and walking exercises. However, detailed data on the duration and intensity of each activity were not available in the dataset, which may limit the interpretation of the association between physical activity and clinical outcomes.

### 2.4. Outcome Measures

The primary outcome measure was all-cause mortality assessed up to 5 years post-diagnosis, based on data from the national death registry. Survival status was ascertained using death registry records within the NHIS database, which comprehensively captures vital status through linkage with national death certificates. Cause-specific mortality data were not available.

### 2.5. Statistical Analysis

Categorical variables were presented as frequencies and percentages, whereas continuous variables were expressed as a median with interquartile range (IQR) due to non-normal distribution. The distribution of continuous variables was assessed using the Shapiro–Wilk test. As most continuous variables were not normally distributed, differences between groups were analyzed using the Mann–Whitney U test for continuous and ordinal variables. Categorical variables were compared using Chi-square or Fisher’s exact tests, as appropriate. Mortality rates were assessed as the primary outcome using all-cause mortality within a 5-year follow-up period after cancer diagnosis, based on linkage to the national death registry. Odds ratios (ORs) with 95% confidence intervals (CIs) were estimated using multivariate logistic regression to evaluate the association between depression/anxiety and mortality. Variables included in the multivariate model were those found to be significantly different between groups in the univariate analysis, including age, regional residence, and economic status. Healthcare utilization variables were not included in the multivariate model, as they were considered potential mediators rather than confounders in the relationship between mental health and mortality. All statistical analyses were performed using IBM SPSS software version 25.0 (IBM Corp., Armonk, NY, USA). A *p*-value of less than 0.05 was considered statistically significant.

## 3. Results

### 3.1. Patient Characteristics

Baseline characteristics were compared between patients with and without post-diagnosis depression or anxiety disorders. Among them, 784 patients (0.9%) were newly diagnosed with either depression or anxiety disorder within one year following their cancer diagnosis. Specifically, 612 patients (0.7%) were diagnosed with depression and 172 patients (0.2%) with anxiety disorders. Table 1 summarizes the demographic and clinical characteristics of the study population. Compared to the control group—defined as patients with gynecologic cancer who were not diagnosed with depression or anxiety disorders within one year after cancer diagnosis—the depression/anxiety group was significantly older (mean age: 67.4 ± 13.0 vs. 64.5 ± 13.6 years; *p* < 0.001). Additionally, this group had a significantly lower average height (155.1 cm vs. 155.8 cm; *p* = 0.002), larger waist circumference (80.0 cm vs. 78.4 cm; *p* < 0.001), and higher AST (SGOT) levels (26.3 vs. 24.7; *p* = 0.010), higher Gamma-glutamyl transferase (GGT) levels (31.4 vs. 25.0; *p* < 0.001), and higher total cholesterol (202.0 mg/dL vs. 198.7 mg/dL; *p* = 0.029) and triglyceride levels (141.8 mg/dL vs. 123.3 mg/dL; *p* < 0.001).

Notably, the cognitive function assessment scores were significantly lower in the control group (6.2 ± 2.0) compared to the depression/anxiety group (7.3 ± 2.78; *p* < 0.001), indicating that patients with post-diagnosis mental health disorders exhibited worse cognitive performance. No significant differences were observed in body weight, BMI, blood pressure, ALT (SGPT), hemoglobin, fasting blood glucose, HDL cholesterol, LDL cholesterol, serum creatinine, or estimated glomerular filtration rate (eGFR) between groups.

### 3.2. Regional and Economic Factors

Significant regional differences were observed, with a higher prevalence of depressive disorders among patients residing in metropolitan areas such as Seoul (19.3%) and Busan (8.9%) compared to other regions (*p* < 0.001). Additionally, the depressive group exhibited significantly lower economic status, as indicated by the higher proportion receiving medical aid and lower quintiles of insurance premiums (*p* < 0.001).

### 3.3. Healthcare Utilization

Healthcare utilization was defined as the use of inpatient and surgical services following cancer diagnosis, specifically measured by the number of hospitalization days and the receipt of gynecologic cancer-related surgical procedures. Patients in the depression/anxiety group exhibited significantly longer hospital stays (median: 19.5 vs. 2.4 days; *p* < 0.001) and were less likely to undergo surgery (0.1% vs. 2.2%; *p* < 0.001), suggesting a higher burden of healthcare use and potential barriers to curative treatment. Although detailed regional and age-stratified analyses were not performed due to dataset constraints, regional differences and age-related trends were considered in the multivariate model.

These findings highlight the need to incorporate mental health assessment into routine gynecologic cancer care to mitigate the elevated mortality risk associated with depression and anxiety.

### 3.4. Mortality Outcomes

Overall, mortality was significantly higher among patients with depression or anxiety disorders (38.4%) compared to controls without these diagnoses (29.9%; *p* < 0.001) (Table 2). Patients diagnosed specifically with depression showed a mortality rate of 38.3%, whereas those with anxiety disorders had a mortality rate of 39.1%, indicating similarly increased mortality risks compared to controls. It should be noted that these mortality outcomes reflect crude all-cause mortality without adjustment for tumor stage or age.

Multivariate analysis revealed that the odds ratio (OR) for mortality was 1.47 (95% CI: 1.31–1.65) for the combined depression/anxiety group, indicating approximately a 47% higher risk of mortality compared to patients without these diagnoses (Figure 2). When analyzed separately, patients with depression had an OR of 1.46 (95% CI: 1.27–1.66), and those with anxiety disorders had an OR of 0.97 (95% CI: 0.74–1.27), suggesting depression as a stronger independent risk factor for mortality than anxiety disorders, although the difference between depression and anxiety disorders was not statistically significant.

## 4. Discussion

This nationwide population-based cohort study provides significant evidence that depression and anxiety disorders are independently associated with increased mortality risk in women diagnosed with gynecologic cancers. Our findings underscore the prognostic importance of psychological well-being in oncology, emphasizing the need for integrated mental healthcare in managing these malignancies.

One key finding of this study is that depression alone significantly increased the risk of mortality in patients with gynecologic cancers (OR 1.46, 95% CI: 1.27–1.66). This is consistent with previous studies conducted in breast cancer populations, where depression was strongly associated with increased mortality risks [17]. Similarly, findings from a nationwide cohort study on patients with inflammatory bowel disease (IBD) demonstrated that coexisting depression significantly worsens clinical outcomes. While the disease context differs, this underscores a broader pattern where mental health disorders adversely affect the trajectory of chronic diseases, including cancer [18]. Such parallels suggest that depression universally exacerbates outcomes across different chronic conditions, potentially through mechanisms involving compromised immune function, increased inflammatory responses, poorer adherence to treatment protocols, and reduced engagement in health-promoting behaviors [19].

Additionally, our study found that anxiety disorders, despite being associated with a slightly lower mortality risk than depression alone (OR 0.97, 95% CI: 0.74–1.27), significantly affected outcomes when co-occurring with depression. This cumulative negative impact of combined mental health disorders aligns with prior findings in breast cancer and IBD, emphasizing a compounded adverse effect when anxiety and depression coexist [20]. Anxiety disorders may aggravate physiological stress responses, leading to chronic activation of the hypothalamic–pituitary–adrenal (HPA) axis and altered cortisol secretion patterns, which have been linked to accelerated tumor progression and impaired immune surveillance [21].

Our findings also highlight significant regional and socioeconomic disparities in the prevalence of depression and anxiety disorders among women with gynecologic cancers. Higher prevalence rates in metropolitan regions such as Seoul and Busan suggest that urban stressors, lifestyle differences, and accessibility to mental health services might play crucial roles in shaping these disparities. Moreover, lower socioeconomic status, reflected by the higher proportion of medical aid recipients in the depression/anxiety group, underscores the impact of socioeconomic factors on mental health, further complicating cancer prognosis and management.

The healthcare utilization patterns observed in our study further illustrate the burdens associated with depression and anxiety disorders. Patients with these mental health conditions experienced significantly longer hospitalizations and fewer surgical interventions, reflecting a potential indirect pathway through which psychological disorders impact cancer outcomes by reducing patient eligibility for curative surgeries due to poor physical health status or compromised compliance with preoperative requirements [22]. These findings parallel observations from breast cancer research, where psychological distress adversely influenced adherence to treatment protocols, ultimately impacting survival outcomes [23].

From a clinical perspective, our study emphasizes the necessity of integrating psychosocial support and systematic screening for depression and anxiety within standard gynecologic oncology care. Previous literature consistently supports the beneficial effects of psychological interventions and antidepressant treatments in improving quality of life and potentially enhancing survival rates among cancer patients [24,25,26]. Additionally, recent studies in breast and pelvic cancer survivors further underscore the importance of addressing patients’ emotional and psychological well-being. For example, Carney et al. reported that many breast and pelvic cancer patients experience persistent psychosexual distress even after completing medical treatment, and that they strongly prefer tailored psychosexual interventions to manage these unmet needs [27]. Hummel et al. demonstrated that breast cancer survivors with impaired sexual function frequently experience comorbid emotional distress, including depression and relationship strain, further supporting the critical link between mood and sexual health [28]. Similarly, Delgado-Enciso et al. found that breast cancer survivors with anxiety had over 11 times greater odds of experiencing sexual dysfunction compared to those without anxiety, emphasizing the psychological dimension of post-cancer morbidity [29]. Although our current data did not directly evaluate therapeutic interventions, the robust association between mental health disorders and increased mortality strongly advocates for routine psychological assessment and timely mental health interventions to potentially mitigate negative outcomes.

This study’s strengths include its large sample size derived from comprehensive nationwide insurance data, providing high statistical power and generalizability of findings. However, several limitations should be noted. First, its retrospective observational design limits our ability to draw causal inferences regarding the relationship between depression, anxiety, and mortality. Second, the NHIS database lacks detailed clinical variables such as tumor stage, histologic subtype, treatment regimens, and comorbidities. The absence of these factors precludes adjustment for important confounders and restricts our ability to perform stratified analyses by cancer severity or treatment intensity. Third, although regional residence and economic status were included in the multivariate models, we were unable to conduct stratified analyses by geographic or sociodemographic subgroups due to limitations in dataset granularity and statistical power. Environmental and healthcare access-related factors, which are likely to influence mental health and survival outcomes, warrant further exploration in future studies. Fourth, mortality was assessed as all-cause mortality, and cause-specific death data were unavailable, limiting the interpretation of whether deaths were cancer-related or due to other conditions. Lastly, mental health conditions were identified based on ICD-10 diagnostic codes, which may underestimate true prevalence due to underdiagnosis and the sociocultural stigma surrounding mental illness in Korea.

Although the dataset used in this study covers the period from 2007 to 2014, the relevance of our findings remains applicable to current clinical practice. Mental health screening and integrated psychosocial care in oncology remain under-implemented in many healthcare systems, including in Korea. Furthermore, depression and anxiety continue to be prevalent and underdiagnosed among cancer patients, reinforcing the importance of systematic screening and intervention, as supported by our findings.

## 5. Conclusions

In conclusion, depression and anxiety disorders significantly predict increased mortality in women with gynecologic cancers, particularly when these conditions coexist. In this large-scale national cohort of women with gynecologic cancers, post-diagnosis depression and co-occurring anxiety were significantly associated with increased all-cause mortality. These findings support the integration of routine psychological screening and timely intervention into gynecologic oncology practice to improve not only emotional well-being but also survival outcomes. Further research should explore the efficacy of targeted psychological interventions and antidepressant treatments in reducing mortality risk, guiding comprehensive care strategies for women facing gynecologic malignancies.

## Figures and Tables

**Figure 1 healthcare-13-01904-f001:**
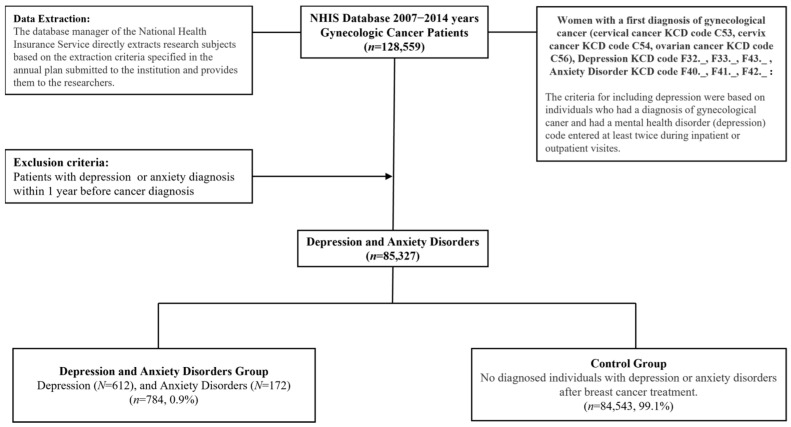
Data cleansing flow chart.

**Figure 2 healthcare-13-01904-f002:**
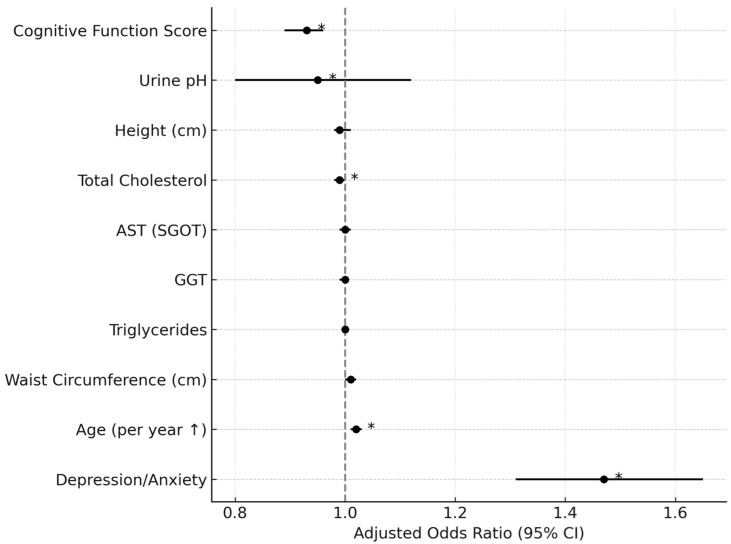
Adjusted odds ratios for mortality by clinical and psychological factors. Values represent unadjusted all-cause mortality. Standardization by age or cancer stage was not possible due to data limitations. Variables marked with an asterisk (*) are statistically significant. The symbol (↑) denotes that the risk of mortality increases incrementally with each additional year of age.

**Table 1 healthcare-13-01904-t001:** Demographic and clinical characteristics of patients with gynecologic cancers, stratified by presence of depression or anxiety disorders following cancer diagnosis.

Characteristics	Patients with Post-Diagnosis Depression/Anxiety Group (*n* = 784)	Patients Without Post-Diagnosis Control Group (*n* = 84,543)	*p*-Value
Age (years)	67.4 (59–77)	64.5 (56–75)	<0.001
Height (cm)	155.1 ± 6.0	155.8 ± 6.1	0.002
Waist circumference (cm)	80.0 ± 9.3	78.4 ± 9.6	<0.001
AST (SGOT)	26.3 ± 18.3	24.7 ± 18.0	0.010
GGT	31.4 ± 42.7	25.0 ± 32.7	<0.001
Urine pH	6.17 ± 0.6	6.08 ± 0.6	0.038
Total cholesterol	202.0 ± 40.3	198.7 ± 42.2	0.029
Triglycerides	141.8 ± 91.1	123.3 ± 80.3	<0.001
Cognitive function score	7.3 ± 2.8	6.2 ± 2.0	<0.001

Data are presented as median (interquartile range). *p*-values were calculated using the Mann–Whitney U test. *p* < 0.05 was considered statistically significant.

**Table 2 healthcare-13-01904-t002:** Mortality outcomes among patients with gynecologic cancers according to the presence of depression or anxiety disorders.

Group	Death, *n* (%)	Alive, *n* (%)	OR (95% CI)	*p*-Value
Depression/Anxiety (*n* = 784)	301 (38.4%)	483 (61.6%)	1.47 (1.31–1.65)	<0.001
Control (*n* = 84,543)	25,279 (29.9%)	59,264 (70.1%)	Reference	<0.001

Data are presented as frequency (N) and percentage (%). Adjustments in multivariate analyses were made for age, height, waist circumference, AST (SGOT), GGT, urine pH, total cholesterol, triglycerides, and cognitive function score. The odds ratio indicates an increased risk of mortality compared to the control group without these diagnoses. Values represent unadjusted all-cause mortality. Standardization by age or cancer stage was not possible due to data limitations.

## Data Availability

The data presented in this study are available from the Korean National Health Insurance Service (NHIS). Restrictions apply to the availability of these data, which were used under license for the current study, and so are not publicly available. Data access may be possible for researchers who meet the criteria for access to confidential data from the NHIS.

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
