# Peer review of "Impact of Depression and/or Anxiety on Mortality in Women with Gynecologic Cancers: A Nationwide Retrospective Cohort Study"

_healthcare, 2025, doi:10.3390/healthcare13151904_

Round 1

Reviewer 1 Report

Comments and Suggestions for Authors

Please find attached a detailed document containing my comments and suggestions regarding the manuscript titled "Impact of Depression and Anxiety on Mortality in Women with 2 Gynecologic Cancers: A Nationwide Retrospective Cohort Study" . The comments are organized by section for clarity and ease of reference.

In summary, the main points include:

  • Clarification needed regarding inclusion of control groups and demographic details.

  • Consistency issues between data presentation (parametric vs non-parametric) and statistical methods used.

  • Clarification and correction of discrepancies between text and tables.

  • Requests for clearer interpretation of results, including odds ratios and confidence intervals.

  • Suggestions to improve the discussion section by incorporating demographic and environmental factors.

  • Recommendations to revise the manuscript for English language clarity, preferably with a native speaker.

  • Need to address the handling of anxiety and depression as separate or combined variables and align the title accordingly.

  • Additional considerations about the verification of causes of death and methodological clarifications.

I trust these comments will be helpful in strengthening the manuscript.

Best regards,

Comments on the Quality of English Language

The manuscript would benefit from professional English language editing to improve grammar, clarity, and overall readability. I recommend seeking support from a native English speaker or a professional language editing service. 

Author Response

Thank you for providing the reviewers’ comments in PDF format.
We have responded directly within the PDF file. Thank you once again for your kind review.

Reviewer 2 Report

Comments and Suggestions for Authors

Review of the Article

Introduction

  1. Remove the word "systematically" from the study aim to avoid confusion with "systematic review" methodology.

Methods

  1. Please define all abbreviations at first mention and maintain consistent usage throughout. Currently, some abbreviations lack definitions while others are redundantly redefined.
  2. The sample size information appears redundantly in both Methods (lines 75-77) and Results (lines 110-112). Please retain this only in the Methods section.
  3. Line 92: Specify the cognitive function assessment tool(s) used, including scoring interpretation. The clinical meaning of scores depends on the specific instrument.
  4. Clarify the mortality assessment timeframe (e.g., 1-year, 3-year, or 5-year post-diagnosis). Without this, the mortality results lack context.

Results

  1. Table 2 and Figure 2 cannot be meaningfully interpreted without standardization of mortality measures.
  2. The interpretation in lines 163-166 belongs in the Discussion section, not Results.
  3. Present regional, economic, and healthcare utilization factors before mortality outcomes, as they are potential confounders for the primary aim (depression/anxiety-mortality association). Justify their exclusion from multivariate modeling.

Discussion

  1. Address how data from 2007-2014 apply to current clinical practice, given potential temporal changes in a decade.
  2. The IBD discussion (lines 176-179) appears to serve primarily for self-citation. While self-citation is acceptable when relevant, ensure it directly supports the current study's context.
  3. The conclusion about "integrating psychological screening in oncology" is overly generic. Conclusions should specifically reflect your findings about depression/anxiety-mortality associations in the studied population to justify the study's purpose.

Author Response

We sincerely thank Reviewer 2 for the valuable comments.
We have provided detailed responses in the "Response to Reviewer" format and revised the main manuscript accordingly.

Round 2

Reviewer 1 Report

Comments and Suggestions for Authors

The authors have addressed all the comments, and the article can be accepted in its current version.

Reviewer 2 Report

Comments and Suggestions for Authors

No additional changes are required.